
# Tangible and intangible ex-post assessment of flood-induced damages to cultural heritage

Claudia De Lucia, Michele Amaddii, Chiara Arrighi

Department of Civil and Environmental Engineering, Università degli Studi di Firenze, Via di S. Marta 3, 50139, Florence, Italy

*Correspondence to:* Chiara Arrighi (chiara.arrighi@unifi.it)

**Abstract.** Floods pose significant risks to cultural heritage (CH), yet post-disaster damage data to CH remain lacking. In this paper, we address this gap by focusing on the ex-post assessment of flood-induced damage to CH. The method involves the identification of damaged assets, and a field survey to assess tangible (LTV) and intangible (LIV) damage. The potential contributing factors e.g., water depth and river slope, are analyzed through geospatial analysis. Ex-post damage data to CH are compared with the outcome of an ex-ante analysis based on available methods to verify the quality of exposed data and possible limitations. The method is applied to the 15-16 September 2022 flood event that occurred in the Marche Region (Italy). The survey involved 14 CH in 4 municipalities and 3 catchments. Results highlight the inadequacy of existing exposure data for ex-ante damage assessment. However, ex-post data confirm that religious architectures are likely to suffer the highest LTV and LIV. The ex-post damage analysis provided a semi-quantitative evaluation of both LTV and LIV in relation to flood characteristics. Notably, significant correlations between LTV and flood depth, as well as with the slope of the riverbed (a proxy for river flow velocity), were found. LIV correlates well to flood depth and river slope although with lower $R^2$ and larger RMSE, highlighting that intangible impact analysis requires more effort than hazard characterization. Further research should increase the availability of ex-post damage data to CH to pose the basis for damage model validation and development of empirical vulnerability functions.

## 1 Introduction

Floods are among the most frequent and costliest natural hazards (CRED, 2015; Wallemacq et al., 2015). In recent decades, the frequency and intensity of heavy rainfall, associated with ongoing climate change have consequently led to an increase in flood events (Merz et al., 2021; IPCC, 2023). Moreover, due to severe urbanization and increasing development in flood-prone areas, flood impacts are expected to grow in the future (Dottori et al., 2023).

The EU Flood Directive calls upon member countries to mitigate the potential adverse consequences of flooding on human health, the environment, cultural heritage, and economic activities (EU Flood Directive, 2007/60/EC). Concerning cultural heritage (CH), this purpose gains even more significance. Indeed, CH assets are severely affected by floods and are likely to be increasingly threatened by climate change effects (Marzeion and Levermann, 2014; Fatoric and Seekamp, 2017; Sesana et al., 2021). In many cases, substantial costs for restoration are necessary, and in the worst-case scenario, the irreversible destruction of unique and irreplaceable assets that hold cultural significance is unavoidable (Arrighi, 2021; Arrighi et al., 2023b). Furthermore, the impact of floods on CH extends beyond the tangible damage, affecting social identity and cohesion (Romão et al., 2020).

Cultural heritage can be defined as the legacy of tangible and intangible attributes inherited from past generations. Tangible attributes include buildings, monuments, and historic places, as well as works of art, literature, music, and artifacts both archaeological and historical. Intangible attributes comprise social customs, traditions, and practices, rooted in aesthetic and spiritual beliefs and oral traditions (Willis, 2014).



Over the past decades, ex-ante damage assessment, namely impact analysis and mitigation measures of natural hazards
to CH assets, such as floods, received considerable scientific attention. Many researchers focused on individual assets or
site levels (Sabbioni et al., 2006; Drdácký, 2010; Huijbregts et al., 2014; Sesana et al., 2021; Momčilović Petronijević
and Petronijević, 2022; Anderson, 2023). Other studies focused on the negative effects of natural hazards on CH
concerning societal impacts and economic losses (Alexandrakis et al., 2019; Garrote et al., 2020). Additionally, several
studies have focused on flood risk assessment of CH at various scales, ranging from specific sites (Garrote et al., 2019;
Zhang et al., 2024), to cities (Wang, 2015; Arrighi et al., 2018; Trizio et al., 2021; Schlumberger et al., 2022; Arrighi et
al., 2023a; Brokerhof et al., 2023; Ravan et al., 2023), regions (Godfrey et al., 2015; Figueiredo et al., 2020; Arrighi et
al., 2023b), national levels (Stephenson and D'Ayala, 2014), and even globally (Reimann et al., 2018; Arrighi, 2021).
The ex-ante analyses represent a key aspect of any "flood risk management plan", as required by the EU Flood Directive
(EU Flood Directive, 2007/60/EC). However, estimating the loss after an event is equally important to support emergency
management and decide priorities for reconstruction and victim compensation (Molinari et al., 2014). Furthermore,
identifying key factors influencing the vulnerability of CH assets is necessary for a more robust risk assessment.
Achieving this requires the availability of post-disaster loss information and data, coupled with appropriate ex-post
damage analyses. Such endeavours would highlight weaknesses in current risk management practices and thus improve
the effectiveness of preparedness and resilience strategies (Arrighi et al., 2022). Nevertheless, there are only a few
examples in the literature concerning the ex-post assessment of damage to CH. In the work of Vecvagars (2006), an
overview of the different available methods in assessing the value of CH assets, providing some recommendations for
valuing damages and losses after a disaster, is outlined. Since 2008, the European Commission, the United Nations
Development Group, and the World Bank developed the joint Post-Disaster Needs Assessment (PDNA) tool. This tool
provides a comprehensive, government-led assessment of post-disaster damages, losses, and recovery needs, paving the
way for a consolidated recovery framework. The PDNA framework encompasses the gathering of data on damages to
both tangible and intangible values of cultural assets. More recently, a reviewed version of the PDNA, based on
experiences gathered through the analysis of many PDNA post-disaster assessments conducted since 2008, was published
(Jeggle and Boggero, 2018). Vafadari (2017) developed a tool for the recording and inventory of sites and monuments as
well as to record damage and threats, their causes, and assess their magnitude. Deschaux (2017) details the observed
impacts on movable and immovable heritage following the floods in Central France in 2016. Figueiredo et al. (2021)
analyze the impacts of wildfires that occurred in Portugal on cultural heritage integrating geospatial analysis with
information provided directly by municipalities affected by the wildfires.
As already mentioned, CH assets are characterized by both tangible and intangible value, and consequently, the damage
they suffer can be tangible and intangible. Therefore, for an adequate assessment of flood damage to CH, a classification
of these values is necessary (Romao et al., 2020), whether the analysis is conducted ex-ante or ex-post. Vecvagars (2006)
groups cultural heritage values into "use value" (related to market value) and "non-use value" (i.e., non-market value such
as spiritual value, legacy value, and social value). In addition, use value can be further divided into "extractive use value"
and "non-extractive use value". Extractive use value includes consumptive value, which can be measured through market
transactions. Non-extractive use value originates from the service the asset provides and includes aesthetic and
recreational values.
However, it is often noted that quantitative disaster data concerning losses related to cultural heritage are either scarce or
entirely unavailable (Romão et al., 2020). This underscores the persistent challenges in obtaining comprehensive
information on the impact of disasters on cultural heritage, emphasizing the need for improved data collection and
assessment methodologies in this critical domain, which are essential for damage model calibration and validation.


This paper focuses on the analysis of damage to CH assets as a consequence of a flood event. First, an ex-ante analysis
was performed using the available data. The official existing hydraulic hazard maps and the national CH database were
considered. However, the pivotal aspect of this study lies in the ex-post damage assessment. A well-defined workflow
has been proposed to assess the tangible and intangible losses incurred by CH due to flooding: (i) identification of the
assets potentially damaged by the flood; (ii) field data collection for the assessment of damage to CH; (iii) ex-post damage
assessment considering both tangible and intangible values of the damaged assets; (iv) analysis of the possible
contributing factor of the damage to CH.
The proposed method is applied to the case study of the flood event that interested the Marche Region (Central Italy) on
15-16 September 2022. The involved sites encompass different types of assets such as churches, historic bridges, and
industrial buildings, which are located in three basins in the Marche region: Burano, Cesano, and Misa.
Through the method proposed in this paper, we aim to fill the gap in the literature concerning ex-post assessment of
cultural heritage damage induced by floods. The research pinpoints the factors that significantly contribute to the
vulnerability of cultural heritage and the resulting flood-related damages.

## 2  Materials and methods

This section outlines the evaluation of flood damage to CH assets using two approaches: ex-ante and ex-post. Sect. 2.1
details the ex-ante damage analysis, which was conducted using available data. On the other hand, Sect. 2.2, the focus of
the paper, describes the procedure for the ex-post damage assessment.

### 2.1  Ex-ante damage assessment

The aim of the ex-ante damage assessment is to investigate if using the available data before the flood event, it would
have been possible to predict the degree of flood damage to CH. The database of CH considered for this analysis consists
of the assets included in the national MIC database (Istituto Superiore per la Conservazione ed il Restauro – MiBACT,
2024). The database contains movable and immovable assets under protection with declared cultural interest. In addition,
it includes assets older than 50 or 70 years under evaluation to verify their effective cultural interest (D.lgs. 22 gennaio
2004, n. 42). The assets that overlap with the official map of flood hazard areas are then considered. The ex-ante damage
assessment was evaluated as the combination of exposure and vulnerability (Arrighi et al., 2023b).
Exposure of CH can be evaluated by intersecting the shapefile of CH with the official flood hazard map available from
the website of the competent authority (AUBAC, 2024).  As the MIC database does not provide information about the
value of the assets and only contains items of national listing, an exposure score equal to 1 ($E = 1$) is assigned to all
assets that overlay areas with some probability of inundation (i.e., P3 – high probability; P2 – medium probability; P1 –
low probability). On the other hand, a 0 score is attributed to all those assets that are not potentially flooded.
According to the vulnerability classification of Arrighi et al. (2023b), a vulnerability class is defined for each CH based
on its typology.
-    Very high vulnerability: religious, residential, tertiary, fortified architectures, and museums.
-    High vulnerability: industrial, productive, rural architectures, and monuments.
-    Medium vulnerability: archaeological areas, infrastructure, and plants.
-    Low vulnerability: open spaces.



According to this approach and based on the available data, considering the same value (E=1) for all assets then results
in damage equal to vulnerability.

### 2.2 Ex-post damage assessment: The workflow

The proposed workflow consists of 4 steps. The first step is focused on the identification of CH assets actually damaged
by the flood (Sect. 2.2.1). Then, in the second step, a post-event field survey, based on on-site visual inspection, is
conducted to evaluate the actual state and condition of CH assets (Sect. 2.2.2). Once all the data and information on the
damage to CH assets is obtained, the ex-post evaluation can be carried out assigning to the assets both tangible and
intangible values and related losses (Sect. 2.2.3). Lastly, the analysis of which factors contributed most to the damage, by
means of geospatial methods, is performed (Sect. 2.2.4).

### 2.2.1 Identification of CH assets potentially damaged by the flood

The initial step is dedicated to identifying CH assets situated within the flooded areas. The CH database of MIC could be
considered for this analysis. The data can be downloaded from the MIC cartographic tool (Istituto Superiore per la
Conservazione ed il Restauro – MiBACT, 2024) and comprehends architectural and archaeological assets, as point
features. After the field survey verification, the list of the assets included in the MIC database could be modified, possibly
adding, and also disregarding some assets, as explained in Sect. 2.2.2. Once the database of CH is obtained, the
identification of the assets potentially damaged by the flood is accomplished through the availability of the map of flooded
areas (shapefile format) that is freely available for download from the Copernicus Emergency Management Service
(COPERNICUS Emergency Management Service – Mapping, 2022). The flood map generation is based on the
acquisition, processing, and analysis, in rapid mode, of satellite imagery and other geospatial raster and vector data
sources. The identification of potentially damaged assets is obtained by overlaying the shapefiles of the flooded area and
the CH database in a GIS environment. In this way, it is possible to obtain a database of CH assets affected by a flood
event, which contains key information, such as name, type, and geo-localization of each individuated asset.

### 2.2.2 Post-event field data collection

In addition to the CH assets identified as described in Sect. 2.2.1, other assets can be considered and then added to the
database based on feedback from local authorities. Indeed, based on the purpose of the work, immovable and movable
assets characterized by aesthetic, historical, testimonial, and municipal value, as well as those with tourist or local
significance identified by local authorities, are considered. On the other hand, the assets listed in the MIC database that
are not mentioned by local authorities and by official tourism websites or have no reviews on major platforms (e.g.,
TripAdvisor and Google), could be excluded.
A novel procedure for data collection aimed at assessing the damage to CH as a result of flooding has been conceptualized.
The data collection forms implemented by Molinari et al. (2014) for residential buildings and industrial facilities were
modified and adapted to the characteristics of CH. Besides the information about the asset, the flood event (e.g., maximum
water level), the presence and typology of any movable artworks, and the observed physical damages, the form allows
for the registration of the cultural value of the CH. Table 1 summarizes the information collected on the field, through the
survey form.



**Table 1 - Survey form: description of CH assets and aspects considered.**

| Form type | Description of CH | Aspects |
|---|---|---|
| General information | | |
| | General features of CH | Geographic coordinates or address<br>CH denomination<br>Level of listing<br>Typology of CH<br>Current use<br>Cultural value<br>Property<br>Fieldworker |
| | Building features | Period of construction<br>Building structure<br>External ornamental elements<br>N° of floors and building height |
| | Description of flood characteristics | Duration<br>Max. water level outside the building<br>Max. water level inside the building<br>Sediments grain size or contaminants |
| | Identification and type of damage | Structural, loss of accessibility<br>Features damaged |
| Building internal damage | | |
| | Damage to floors (exposure/vulnerability of the containing building) | Covered and uncovered surface<br>Level of maintenance<br>Presence and type of plants<br>Damage to frescoes and wall paintings, doors and windows, floors, plants |
| Contents damage | | |
| | Identification of movable assets | Presence and type of artworks |
| | Damage to the artworks (exposure/vulnerability of contents) | Damage to furniture, paintings, sculptures, books, decorative items, votive and liturgical elements, textile, archaeological finds |

The form also reports the measure of "Max. water level outside the building" (see the general information form of Table
1). This task refers to the on-site evaluation of the maximum height reached by water around the damaged CH assets as
a result of the river overflow (hereafter MWL). The measurement for each asset can be obtained by taking as a reference
a point along the perimeter where mud marks were still visible at the time of the field campaigns. In case variations in the
maximum level are evident around the perimeter of the building, multiple measurement points can be noted, and then the
average height value can be determined. The height of the mud marks from the ground can be measured with the use of
a classical meter and/or laser distance meter. The chosen reference level (i.e., the relative level of 0 for each asset) is
usually in a flat area whose coordinates will be easily found for subsequent office analysis. Indeed, referring to the
measurement of the maximum height reached by the water to a reference level is necessary to correctly locate the data
collected on a GIS system. Therefore, the difference in height between the maximum level reached by the water outside
the considered CH asset and the reference level is measured. Practically, an operator, using a laser distance meter, points



horizontally from the measurement level to the reference level, while another field operator located on the reference point,
can measure the height of the laser from the ground level. In the case of a bridge, the reference level from which the
MWL is measured corresponds to the deck.
As concerns the cultural value assignment, the following procedure is adopted. Based on the qualitative descriptors
introduced by Historic England (HE, 2008), non-extractive and non-use values were outlined in four categories:
evidential, historical, aesthetic, and communal value:

- ● Aesthetic value: includes aspects of sensory and intellectual stimulation from the CH.
- ● Historical value: derives from the connection between the past and the present through the asset. It includes (i)
- illustrative value if the asset illustrates something unique or rare and (ii) associative value if it is associated with
- a notable family, person, or event.
- ● Evidential value: derives from the potential of the asset to yield evidence about past human activity.
- ● Communal value: derives from the meanings of a place for the people who relate to it, or for whom it figures in
- their collective experience or memory. It encompasses (i) commemorative value, (ii) social value, and (iii)
- spiritual value.

Each category of value can be described by four qualitative levels ranging from unknown to high value: the respective
"$V$" score was assigned to each asset. It is noteworthy that the chosen hierarchical system excludes the "no value" level,
preferring to assign an "unknown value" to the asset without evidence that would support its significance. Table 2
summarizes, for each category of value, the criteria to be considered when assessing the level of value of the cultural
property and the scores corresponding to each class of value.
**Table 2 - Classification and criteria to define intangible value of CH with their respective class and associated score.**

| Type of value | Criteria to assign CH value | Class value and score ($V$) |
|---|---|---|
| Aesthetic | Valuable structure (e.g., architectural art using local materials or high-value import materials); valuable artworks inside (objects of outstanding workmanship, precious votive elements) | High (10) |
| | Valuable structure or valuable artworks | Moderate (7) |
| | No uncommonly attractive qualities, but that display particular characteristics of an identified style | Limited (3) |
| | No valuable characteristics or stylistic features | Unknown (0) |
| Historical | Proved illustrative and associative value or pre-1800 structure | High (10) |
| | 1800 structure | Moderate (7) |
| | 1900 structure | Limited (3) |
| | Structures under 70 years of age | Unknown (0) |
| Evidential | Physical remains of past human activities. The current use has not deleted proofs of the past | High (10) |
| | No evidence of the past, but their history is based on past human activity | Moderate (7) |
| | Only the denomination recalls past human activity | Limited (3) |





| | No linked to past human activities | Unknown (0) |
|---|---|---|
| Communal | Spiritual, social, or commemorative value. Additionally, committees have been founded to promote or defend the asset, or the asset is linked to a specific local tradition | High (10) |
| | Spiritual, social, or commemorative value. No committees or traditional events are linked to the asset | Moderate (7) |
| | Limited spiritual value (e.g., place of worship with sporadic openings). No traditional events are linked to the CH | Limited (3) |
| | No spiritual, social or commemorative value | Unknown (0) |

Following Romao and Paupério (2021), the baseline pre-disaster intangible value $BV$ of a certain CH asset will then
correspond to the sum of the scores established for each type of value given by:

$$BV = \sum_{i=1}^{4} V_i \qquad \text{Eq. 1}$$

where $V_i$ is the score of ith typologies of value.
### 2.2.3    Ex-post damage assessment
The level of damage is obtained by combining loss in tangible and in intangible values. Loss in tangible value is strictly
linked to the observed physical damages and to the costs of restoration. It includes structural and non-structural damage.
The Italian Civil Protection Department defines structural damage as those involving the load-bearing elements of the
building, such as pillars, beams, and slabs. In case of non-structural damage, the elements that do not affect the stability
of the building such as ceiling and floor finishes, plumbing, and electrical systems are affected.
On the other hand, loss in intangible value is established by evaluating flood indirect impacts. Loss in aesthetic value
refers to the effectiveness of restoration in allowing the community to be sensorial stimulated by the asset again. The
impact on historical and evidential values depends on how the flood impacted the original structure and materials or the
proofs of past human activities, such as plaques or archives. Finally, the loss in communal value is measurable as the
duration of inaccessibility of the asset (HE, 2008). In this paper, we assume that an asset sustaining moderate damage
may be closed for days or weeks for clean-up and safety check operations, whereas an asset with severe damage may be
closed for months for restoration works. It is also assumed that if an asset remains inaccessible for more than one year
the loss in intangible value is comparable to the destruction, as the community will move to a new place to express
communal value.
Damage is categorized into four hierarchical classes, with each asset assigned both a loss in tangible value (LTV) and a
loss in intangible value ($LIV$). LTV ranges from 5 to 30, while calculating $LIV$ involves applying the methodology
outlined in Romao and Paupério (2021). This method employs a coefficient ($D$), which spans from 0 to 1, associated with
each class of loss or damage. Then, for each cultural heritage asset, the loss in $LIV$ is defined applying Eq. 2:

$$LIV = \sum_{i=1}^{4} V_i \ x \ D_i \qquad \text{Eq. 2}$$

where $V_i$ represents the score of the category of values. As shown in Table 2, the score of $V$ ranges from 0 to 10, while
the coefficient $D$ (Table 3) could be at most equal to 1, resulting in a $LIV$ score that ranges from 0 to 40. This implies,





therefore, that greater weight is given to *LIV* than to LTV to emphasize the peculiar contribution of intangible aspects to
the loss evaluation. The classes of damage and the criteria adopted to define losses value, considering both tangible and
intangible features, are reported in Table 3.
**Table 3 - Classes of damage and definition of LTV and *LIV*.**

| Classes of damage | LTV | *LIV* |
|---|---|---|
| Undamaged    or slightly damaged | CH can return to its original state with deep cleaning. | The intangible values have not been impacted. The site has never been closed off. |
| | LTV=5 | $D = 0$ |
| Moderately damaged | Slight structural and non-structural damages (door unhinged, appliances damaged, and presence of mold). | Restoration can repair most of the features that provide aesthetic, historic, or evidential value. The site has been closed for days or weeks. |
| | LTV=10 | $D = 0.3$ |
| Severely damaged | Building and artworks damaged (wrecked floor, wall painting, sculptures, paintings, furniture, wooden choir, pipe organ, liturgical supply ruined). | Despite restoration works, the damaged features that hold aesthetic, historical, and evidential significance cannot be fully restored to their original state. The site has been closed for months. |
| | LTV=15 | $D = 0.7$ |
| Destroyed/lost | Asset    destroyed    (the    building materials are not on site anymore). | Lost in significance. The site or its most relevant features are destroyed and/or closed for more than one year. |
| | LTV=30 | $D = 1$ |

### 2.2.4    Factors influencing flood damage

Flood damage to buildings can be caused by several factors, both intrinsic, influenced by the properties of the structure
itself, and extrinsic, influenced by the dynamics of the flood event. In literature, the following factors are typically
considered: intrinsic factors, such as the material of construction, the presence of contents susceptible to flood damage
and with significant cultural value, the existence of possible water communication between the building and the river, the
presence of defence elements, age in years of the building, number of floors, building shape, building orientation in
respect to the water flow, state of conservation of the building, and objects that drag the sheet of water; extrinsic factors
such as maximum water level outside the building, flow velocity, hydrodynamic pressure, flood duration, presence of
sediments, and contaminations (e.g., Smith, 1994; Kreibich and Thieken 2008; Dall'Osso et al., 2009; Dutta et al., 2011;
Galasso et al., 2021; Marin Garcia et al., 2023).
These factors can be directly assessed by means of post-event field survey, or by the interpretation of post-event photos
and videos and can be classified based on the level of difficulty in obtaining them (Marin Garcia et al., 2023).
Additionally, other authors (e.g., Cuca and Barazzetti, 2018; Di Salvo et al., 2018; Kefi et al., 2020; Al-Kindi and Alabri,
2024) also consider some geospatial factors as they could influence buildings damage: difference between the level of the
ground floor of the building and the riverbank, distance between river and building, difference between DTM and filled
DTM, local slope, curvature, topographic wetness index (Beven and Kirkby, 1979), stream power index (Moore et al.,
1991), terrain ruggedness index (Riley et al., 1999), and NDVI.



The relationship between MWL and structural damage is well-known in the literature. For its evaluation, post-event field
survey measurements are necessary (as described in Sect. 2.2.2). On the other hand, the evaluation of the geospatial
factors requires the use of source data in vector (e.g., hydrographic network, and buildings) and raster formats (e.g.,
DEM), which are generally available from national or regional databases. Concerning the DEM spatial resolution, the
degree of damage to buildings could result from small variations of the morphology. For this reason, the use of high-
resolution DEM (cell size ranging between 1×1 m and 5×5 m) is recommended, especially in the case of urban flood
analysis (Mark et al., 2004; Adeyemo et al., 2008; Di Salvo et al., 2018).
Specific procedures using GIS tools are implemented to assess two factors: the minimum distance ($\Delta D$) between a CH
asset and the river, and the elevation difference ($\Delta E$) between the CH asset and the riverbed. First, polygon-type shapefile
of the buildings corresponding to CH assets are identified. For $\Delta D$, the centroid of the building polygons is considered,
with the river network as the reference for distance evaluation. Using the centroid of the buildings and the nearest point
on the hydrographic network, the $\Delta D$ factor is determined automatically with GIS tools (e.g., the Near tool in Analysis
Tools of ESRI$^{TM}$ ArcGIS Pro$^{TM}$). Concerning $\Delta E$, for each building polygon, the DTM is extracted. The elevation
difference between the CH asset and the nearest point feature on the riverbed is then calculated. To refine the riverbed
elevation, a buffer distance around the riverbed can be considered.
Concerning the river slope factor (RS), we assume that the average slope of the riverbed is a reasonable proxy for the
river flow velocity, which is difficult to estimate in the absence of instrumented sections or video recordings during a
flood. Moreover, the slope of the river also influences the transport of sediment and the grain size, which in turn can affect
the degree of damage. Based on our best knowledge, there are no specific recommendations for RS evaluation in the
literature. In this paper, the average slope of 500 m and 1000 m upstream stretch with respect to the assets, is considered.
Regarding the other geospatial factors, these can be evaluated as indicated by the relevant literature cited above. To
evaluate the relationship between each contributing factor and the tangible and intangible losses, the mean and median
values of the area of each CH asset polygon are considered.
## 3    Case study
The method is applied to CH assets damaged by the 15-16 September 2022 flood in the Marche Region. This section
includes an overview of the basins, along with a general description of the municipalities and their historical significance
(Sect. 3.1). Moreover, the dynamics of the intense rainfall event and associated flooding are described in Sect. 3.2.
The geospatial data utilized for the analyses outlined in Sect. 2.2 were sourced from official regional and national
databases. Vector data and the numerical technical map of the Marche Region ("CTR", scale 1:10000) were obtained
freely from the Marche regional cartographic data portal (REGIONE MARCHE, Ambiente, 2023). The LiDAR-derived
DEM, with a spatial resolution of 1 m and vertical accuracy of 0.15 m (comprising both DSM and DTM data), was
acquired following a request to the Italian Government's "Ministero dell'Ambiente e della Sicurezza Energetica" (MASE,
Geoportale Nazionale, 2024). Specifically for the coastal area of Senigallia, a portion of the LiDAR data utilized had a
spatial resolution of 2x2 meters.
### 3.1    Overview of the study areas
The CH assets damaged by the flood are distributed across three basins on the eastern slope of the Central Apennine chain
of the Marche Region, in Central Italy (Figure 1a,b). The basins are drained by their respective main rivers, namely
Burano (a right tributary of the Metauro River), Cesano, and Misa (Figure 1b). The highest peak of the study area, Mt.
Catria (1704 m a.s.l.), is situated at the watershed between the Burano and Cesano basins. The highest peak of the Misa
basin corresponds to Mt. Sassone, reaching an elevation of 826 m a.s.l. (Figure 1b).
The CH assets damaged by the flood are included in the municipalities of Cantiano and Cagli (Burano basin), Pergola
and the hamlet of Bellisio Solfare (Cesano basin), and Senigallia (Misa basin), in Pesaro-Urbino and Ancona provinces.
These localities exhibit diverse historical and cultural attributes. The historical significance of Cantiano and Cagli is
notably linked to the ancient Roman road known as the "Flaminia," which was inaugurated between 223 and 202 B.C.
(Clini et al., 2023). One noteworthy site from the Roman period along the Via Flaminia is the Ponte Grosso bridge,
represented by the white dot between Cantiano and Cagli (Figure 1b).

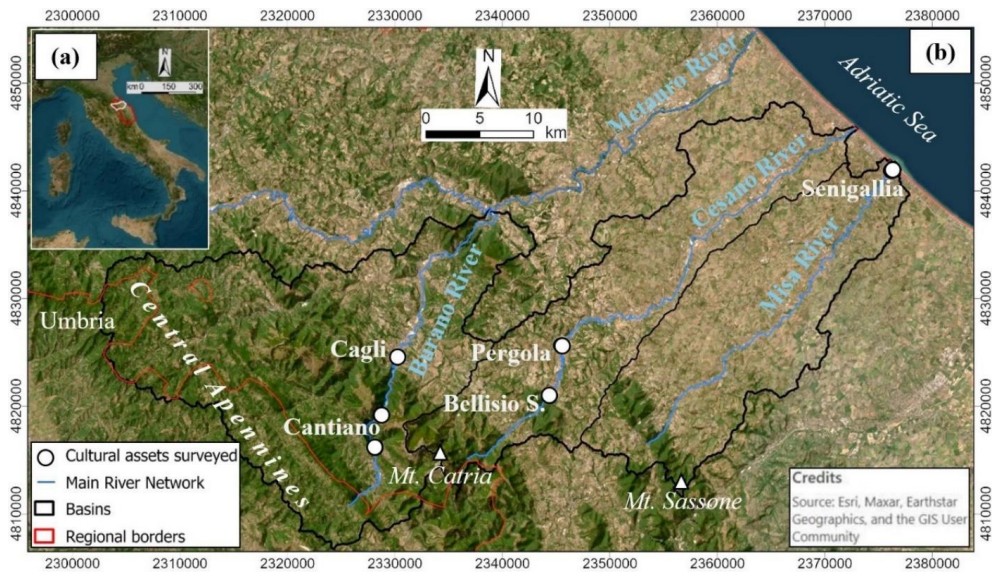


**Figure 1.** (a) The study area in Central Italy. In red is the border of the Marche Region, and in white is the area of the basins which
includes the assets involved during the flood that occurred on 15-16 September 2022. (b) The three basins that include the assets
affected by the flood: Burano, Cesano, and Misa. Coordinate system: WGS 1984 UTM Zone 33N.
As for the Cesano basin, the site of Bellisio Solfare has a recent history starting from the late 1800s, with the beginning
of construction of the sulfur refinery. This location holds significance as part of the Marche Mining Geopark, established
in 2001 (Sulphur, MARCHE MINING GEOPARK, 2024). Pergola, known as the "city of hundred churches", has been
inhabited since Prehistory, with the cultural heritage most extensively documented originating from the Roman period.
The city of Senigallia has a rich historical background, as it was the first Roman colony to settle in the Adriatic coastal
plain. In the realm of flood risk management, the origins of protective measures can be traced back to the early Roman
settlements (De Donatis et al., 2019). Notably, the interventions were directed toward the construction of walls along the
course of the Misa River, with the dual function of both military and flood defense of the Senigallia city. The construction
of the walls, as well as other changes to the minor hydrographic network carried out by the Romans, preserved the city
from flooding by the Misa River. However, during the post-Roman age, the dismantling of these walls exposed a
significant portion of the city to floods, as evidenced by the event in 1472 and subsequent flooding between the 16th and
18th centuries A.D. The aftermath of these post-Roman age flood events, combined with continuous human interventions
contributed to shaping the current topography of the urban area in Senigallia (De Donatis et al., 2012).



### 3.2 The 15-16 September 2022 flood event

On 15-16 September 2022, following an extended period of drought in the preceding months (Pulvirenti et al., 2023), the Northern Marche Region experienced very intense rainfall due to the formation of a stationary self-regenerating thunderstorm system over the Apennine mountains, resulting in disastrous floods. From early afternoon on 15 September, rainfall started to affect the Mt. Catria area, until it also extended to the mountainous areas of the Burano, Cesano, and Misa basins. In Figure 2 the rainfall and hydrometric data of the event are reported. The data were downloaded from the Civil Protection monitoring system website of the Marche region (SIRMIP ON-LINE, 2024) and then elaborated.

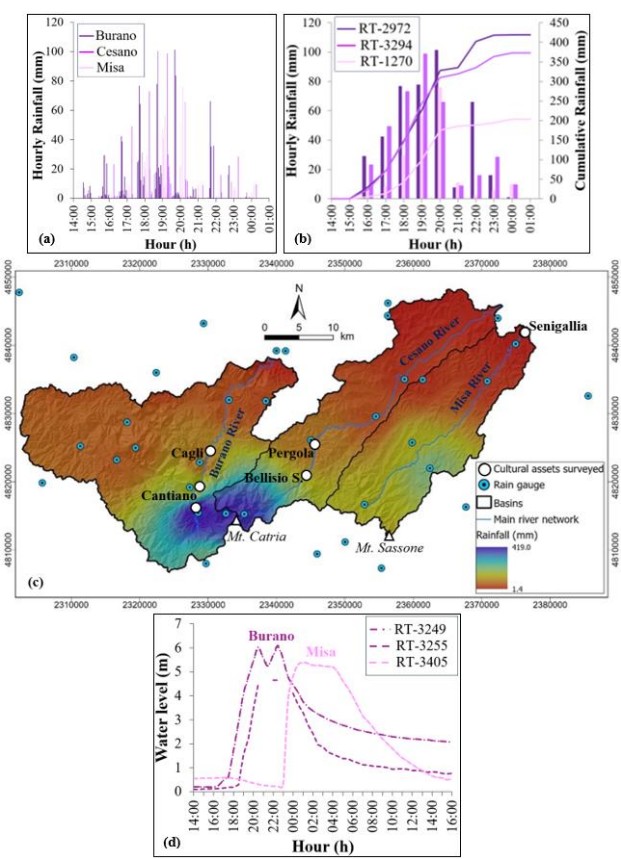

**Figure 2.** Observed rainfall and flow rate of the 15-16 September 2022 event. a) Hourly rainfall measured by the rain gauges in the 3 basins; b) The 3 rain gauges* of each basin that measured the maximum cumulative rainfall; c) Map of the cumulative rainfall; d) Measured water level by hydrometer** of the Burano River and Misa River. *Rain gauges codes: "Cantiano RT-2972" (Burano basin); "Monte Acuto RT-3294 (Cesano basin)"; "Colle RT-1270" (Misa basin). **Hydrometers codes: "Pontedazzo RT-3249" (1 km downstream Cantiano, Burano River) and "Cagli Ponte Cavour RT-3255" (Burano River); "Ponte Garibaldi RT-3405" (Senigallia, Misa River). The shaded relief basemap of panel (c) was obtained from the TINITALY DEM (Tarquini et al., 2007, 2023). Distributed under the CC BY 4.0 license. Coordinate system: WGS 1984 UTM Zone 33N.

The most intense phase of the event occurred between 18:00 and 19:00, with maximum hourly peaks of about 100 mm recorded by stations near Mt. Catria, at the watershed between Burano and Cesano basins. In the Misa basin, the maximum hourly peak was recorded at 19:30, amounting to about 80 mm (Figure 2a,b).





The map of Figure 2c, obtained interpolating the rain gauges data using the inverse distance weight interpolation method
(Shepard, 1968) in ESRI™ ArcGIS Pro™ (IDW tool in Spatial Analyst Tools), highlights the high spatial variability of
the rainfall event.
The rain gauges surrounding Mt. Catria, at the watershed between the Burano and Cesano basins, recorded the highest
hourly rainfall intensity and cumulative rainfall, reaching 420 mm in 12 hours. In contrast, in the Misa basin, the maximum
cumulative rainfall recorded northeast of Mt. Sassone is half the amount that has rained in the Mt. Catria area. In just 6
hours, about half the precipitation that typically occurs on average in a year (i.e., 780 mm, REGIONE MARCHE,
ANNALI IDROLOGICI, 2021) fell in the mountainous areas of the Burano, Cesano, and Misa basins. A return period of
> 1000 years has been estimated for rainfall durations of 3-6-12-24 hours at the rain gauges located in areas characterized
by higher rainfall intensities (REGIONE MARCHE, RAPPORTO DI EVENTO preliminare, 2022).
Although about half as much rain fell in the Misa basin as in the Burano and Cesano basins, the effects were still
disastrous. One reason can be attributed to the different geology of the basins (e.g., Iacobucci et al., 2022). The Mt. Catria
ridge in the Burano and Cesano basins mainly consists of fractured carbonate rocks, that contribute to the infiltration
processes (Mastrorillo and Petitta, 2014), mitigating flood effects. On the other hand, the Misa basin is mainly composed
of clays and sandstones, which are less permeable. As a result, a larger portion of the rainfall contributed to runoff
processes, exacerbating flood dynamics.
The hydrometers reported in Figure 2d, in the Burano basin, are located in the Pontedazzo section which is 1 km
downstream from Cantiano (RT-3249), and in Cagli (RT-3255). The intense rainfall that fell over a brief period led to an
abrupt increase in the river discharge, as highlighted by the water level variations of the Burano and Misa rivers (Figure
2d). The blockage of bridges and culverted stretches significantly contributed to the flooding. In Cantiano, the flooding
of the urban centre occurred from the culverted section of the Burano River, as shown in some videos recorded by
residents (e.g., *World Events News, 2022*). In the case of Senigallia, a video shows the evolution of the flooding of the
Misa River (*Storm Chasers Marche, 2022*). In this case, large woody debris crashed against the deck of the bridges "Corso
2 Giugno" and "Garibaldi" (where the hydrometer is located), causing widespread flooding throughout the city.
A total of 13 people died, and severe damage resulted in most settlements along the main rivers. Further details on flood
dynamics in Cantiano, Cagli, Pergola, and Senigallia, and the consequent damage to CH assets, are provided in Sect. 4.2
of the results.
## 4. Results and discussion
The results of applying the proposed method to assess the damage to CH assets caused by the flood event that occurred
on 15-16 September 2022, in the Burano, Cesano, and Misa basins, are presented and discussed in two main sections.
Sect. 4.1 concerns the analysis of the results obtained by applying the ex-post damage assessment method, which is the
main goal of this paper. In Sect. 4.2 the results of the ex-ante application are compared with the ex-post results and then
discussed.
### 4.1 Ex-post damage assessment
#### 4.1.1 Features of the CH assets and losses assessment



Through remote analysis and field survey verification, the list of CH assets actually damaged by the flood was obtained.
A total of 14 assets were identified, for which, maximum water level (MWL) baseline value ($BV$), and both losses in
intangible ($LIV$) and tangible (LTV) scores are provided in Table 4. Most of the damaged CH assets are religious building
types (6 out of 14), while the remaining damaged assets include bridges, a fortified gate, a square, a porch, and residential
or industrial architecture.
**Table 4 – CH assets damaged by the flood, classified by basin, type, MWL, and the associated scores of $BV$, $LIV$, and LTV.**
**Can: Cantiano; Cag: Cagli; P: Pergola; BS: Bellisio Solfare. All the assets in the Misa basin are located in Senigallia.**

| Basins | CH assets | Type | MWL (m) | $BV$ (-) | $LIV$ (-) | LTV (-) |
|---|---|---|---|---|---|---|
| Burano | (1) S. Emidio oratory (Cag) | Church | 2.40 | 20 | 7 | 10 |
| | (2) Ponte Grosso (Can) | Bridge | 2.50 | 23 | 2.1 | 10 |
| | (3) S. Agostino church (Can) | Church | 0.35 | 27 | 0 | 5 |
| | (4) S. Giovanni Battista collegiate (Can) | Church | 1.40 | 27 | 13 | 15 |
| | (5) S. Nicolò church (Can) | Church | 2.05 | 24 | 5.1 | 10 |
| | (6) Historical buildings Via Fiorucci (Can) | House | 2.30 | 17 | 2.1 | 10 |
| Cesano | (7) S. Maria delle Tinte church (P) | Church | 3.40 | 37 | 20 | 15 |
| | (8) Bellisio Solfare refinery (BS) | Factory | 2.66 | 27 | 27 | 30 |
| Misa | (9) Porta Lambertina | Fortified gate | 0.44 | 17 | 0 | 5 |
| | (10) S. Maria del Porto church | Church | 0.70 | 21 | 0 | 5 |
| | (11) Foro Annonario | Square | 0.65 | 24 | 4.5 | 5 |
| | (12) Portici Ercolani | Porch | 1.50 | 17 | 0 | 5 |
| | (13) Ponte Garibaldi | Bridge | 2.18 | 6 | 6 | 15 |
| | (14) Filanda Serica | Factory | 0.23 | 10 | 0 | 5 |

Figure 3a shows the general view of the basins, and panels b-g highlight the distribution of the $BV$ and $LIV$ scores for the
sites of the three basins, while Figure 4 reports the distribution of the LTV scores throughout the basins (panels b-g);
panels b1-g2 depicts two examples how the MWL was estimated during the field survey, in the case of a generic building
and a bridge, respectively; and in panels b1-c2 are reported two post-event photos showing the MWL.
The most valuable cultural asset corresponds to the S. Maria delle Tinte Church ($BV = 37$), which is located in Pergola,
within the Cesano basin (Figure 3, panel e7). The maximum aesthetic, historical, and communal values are assigned to
that asset, as the church was adorned with statues and stucco decorations, in addition to precious 18th-century wooden
pews, painted with floral motifs. Moreover, the church was built at the behest of the historical dyers and wool merchant
guild, and still today it is a representative place in the city. Indeed, after the 2022 flood, a committee called "Gli Angeli
delle Tinte" was assembled to propose a restoration project for the church (GLI ANGELI DELLE TINTE, 2024). In
general, religious architectures were built before the 1800s and, in addition to the high spiritual value, valuable structures
and valuable artworks coexist, resulting in a high aesthetic value. For these reasons, the average intangible value score of
the damaged churches is relatively high ($BV = 26$), in confront with the average score of the other asset types ($BV =$

363    18).

Ponte Garibaldi (Figure 3a panel g13), namely the damaged bridge in Senigallia (Misa basin), has the lowest intangible
value ($BV = 6$) for its limited historical value (it dates to the 20th century), as well as for its limited aesthetic value.
Indeed, even if it is an example of the typical early 20[th]-century architectural style, it is not a valuable structure. On the
other hand, the other damaged bridge in the Burano basin, Ponte Grosso in Cantiano (Figure 3a, panel 3c), is characterized
by a higher intangible value ($BV = 23$). In this case, even if its aesthetic value is limited, both the historical and evidential
values are high, because it is a rare example of infrastructure of the Ancient Rome Empire.
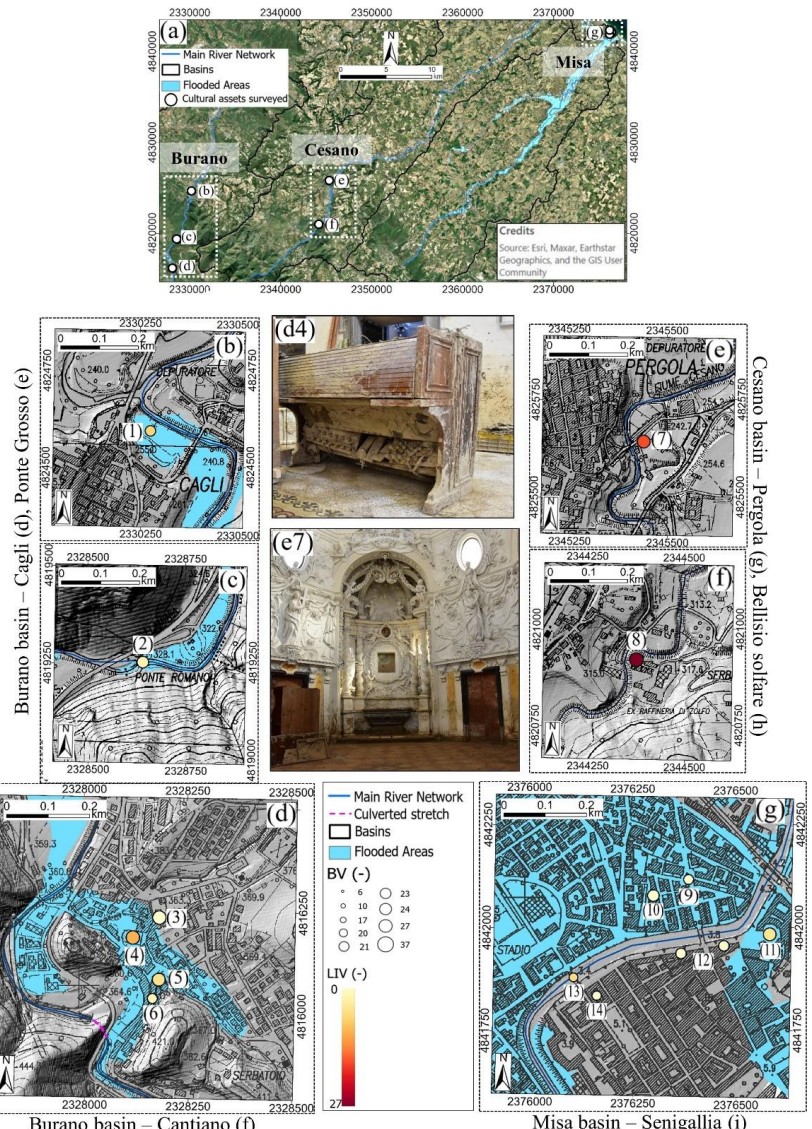

**Figure 3**. (a) General view of the CH assets surveyed for each basin; (b-g) the maps showing the *BV* (graduate symbols) and *LIV* (scale colors) scores of the assets. Burano basin: (b) S. Emidio oratory in Cagli (1), (c) Ponte Grosso in Cantiano (2), and (d) the assets in Cantiano (3-6); Cesano basin: (e) S. Maria delle Tinte Church (7), and (f) Bellisio Solfare (8); Misa basin: (g) the assets in Senigallia (9-14). Panels d4 and e7 report post-event photos of S. Giovanni Battista collegiate and S. Maria delle Tinte church where damage as a result of mud deposition inside the buildings is visible. The shaded relief basemap of panels (b-g) was obtained from the DTM LIDAR of the Ministero dell'Ambiente e della Sicurezza Energetica (MASE, Geoportale Nazionale, 2024). The numerical technical map of panels (b-g) is from the Marche Region (REGIONE MARCHE, Ambiente, 2023). Both maps are distributed under the CC BY 4.0 license. Coordinate system: WGS 1984 UTM Zone 33N.

It is worth noting that the Bellisio Solfare refinery asset (Figure 3a, panel f8), despite being mostly unknown among the most important tourist attractions and with a poor state of conservation, is characterized by high intangible value (*BV* = 27). Indeed, it represented an important proof of the past industrial activity of the Pergola municipality area (Burano basin). Furthermore, a high communal value is assigned to it, due to the presence of an organization that aims to rebuild the asset.



The assets of Historical Buildings Via Fiorucci (Figure 3a, panels d5) and Porta Lambertina (Figure 3a, panel g9) are
distinguished by their high historical significance, being notable architectures of the past, and holding a moderate aesthetic
appeal, resulting in a $BV = 17$. In contrast, Foro Annonario (Figure 3a, panel g11) and Portici Ercolani (Figure 3a, panel
g12), are CH open spaces of notable value, with $BV = 24$, and 17, respectively. While these two assets share similar
evaluations across most value types, the Foro Annonario holds significant community value. Indeed, it represents the
historical central marketplace of Senigallia, thus remaining a vital meeting point for the city since its realization.
Moreover, Figure 3a (panels b-g) reports the extension of the flooded area from the Copernicus agency. In general, these
maps agree with those actually flooded as a result of the event (the same for Figure 4). The only exceptions are the areas
of Pergola and Bellisio Solfare, as well as assets #12,14 in Senigallia. This demonstrates that these maps are useful for
rapid identification of flooded areas. However, a direct field evaluation to establish which assets were effectively flooded
is fundamental.
In Figure 4 are reported the maps showing the spatial distribution of the LTV scores of each asset (panels b-g). Concerning
the Bellisio Solfare refinery (Figure 4, panel f8), the highest LIV and LTV were assigned as the flood destroyed completely
the building, and during the survey, only ruins were observed (LIV = 27, and LTV=30). The historic S. Maria delle Tinte
church (Figure 4, panel e7) sustained considerable damage caused by the flood, both in terms of damage to intangible and
tangible value (LIV = 20, and LTV=15). The inundation resulted in harm to the electricity system and the emergence of
mold on both the floor and wall paintings. Additionally, the force of the floodwater partially wrecked the door and
destroyed the 18th-century pews. As a result, the aesthetic value of the church was deemed lost. Moreover, its extended
closure period led to a significant impact on its communal value. Even the S. Giovanni Battista collegiate (Figure 4, panel
3f) experienced severe damage (LIV = 13, and LTV=15). In addition to the effects already observed for the other assets,
floor tiles were broken, the wooden choir and altars were swollen due to the floodwater, and the 16th-century liturgical
supply was covered by mud. In the case of S. Nicolò church (Figure 4, panel d5), part of the floor collapsed, and the
external stone and metal balustrade were swept away by the flowing water (LIV = 5.1, and LTV=10). Similar loss scores
were observed for the St. Emidio oratory (Figure 4, panel b1), in which, however, a significant loss was due to the wooden
door as it was swept away.
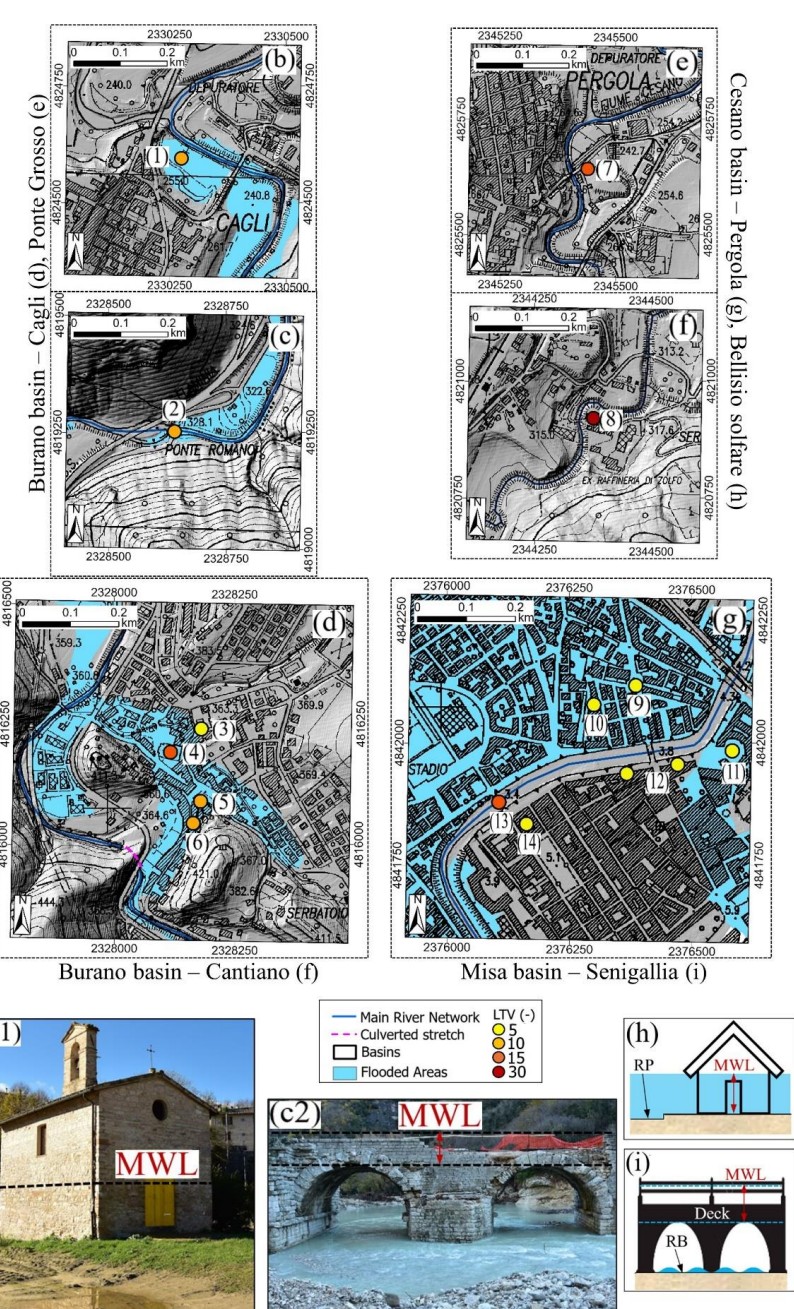

**Figure 4.** (d-i) The maps of the LTV scores of the assets. Panels (b1) and (c2) display the post-event field survey photos depicting the damage to the S. Emidio oratory and Ponte Grosso, respectively. Panels (h) and (i) report the schematic view of the MWL estimation in the case of a generic building and a bridge, respectively (RP is the reference point used for the measurement of the MWL, and RB is the riverbed). The shaded relief basemap of panels (b-g) was obtained from the DTM LIDAR of the Ministero dell'Ambiente e della Sicurezza Energetica (MASE, Geoportale Nazionale, 2024). The numerical technical map of panels (b-g) is from the Marche Region (REGIONE MARCHE, Ambiente, 2023). Both maps are distributed under the CC BY 4.0 license. Coordinate system: WGS 1984 UTM Zone 33N.





Overall, a high level of losses was observed for most of the affected religious structures, where closure due to extensive
damage contributed to a decrease in communal value. Conversely, the S. Agostino (Figure 4, panel d3), and Porta
Lambertina, S. Maria del Porto, Portici Ercolani, and Filanda Serica assets (Figure 4, panels g9,10,12, and 14) incurred
the lowest losses, both in intangible and tangible aspects $LIV = 0$, and LTV=5. Specifically, the two churches were not
damaged as they are over-elevated from the ground floor. For all these assets, only mud marks dirtied the external walls.
As regards the Foro Annonario (Figure 4, panel g11), the only damage is related to the mud marks along the porch
perimeter. Nevertheless, the relative $LIV$ is higher than 0 ($LIV = 4.5$) since the circular square in which the porches are
located remained impracticable for some days.
The two affected bridges were significantly damaged as the maximum level reached by the water during the flood
exceeded the height of the deck. Portions of the arch stones of the Ponte Grosso (Figure 4, panel c2) collapsed leading to
a moderate decrease in tangible value (LTV=10). However, the historical and evidential aspects remained unscathed,
resulting in a relatively low decline in intangible value ($LIV = 2.1$). Conversely, the Ponte Garibaldi (Figure 4, panel
g13) sustained severe structural damage (LTV=15). Indeed, some months after the field survey, it ultimately had to be
demolished (*ANSA, Regione Marche, 2023*), resulting in the loss of both its aesthetic and historical significance ($LIV =$
431   6).
Regarding the MWL estimate (Figure 4, panels h,i), it was directly measured during the field survey, as detailed in Sect.
2.2.2. However, there were exceptions with the two bridges and the Bellisio Solfare refinery. Direct measurements were
not possible in these instances due to the inaccessibility of the bridges, compounded by the destruction of the Bellisio
Solfare asset. Consequently, for these cases, the estimation of MWL was conducted indirectly.
As for the Ponte Grosso (Figure 4, panel c2), the MWL was estimated considering wood deposition height at road signals
close to the bridge (e.g., video from *TGCOM24, 2022*). The resulting estimated MWL from the deck is 2.5 m.
With regards to Ponte Garibaldi (Figure 4, panel g13), the highest water level value from the riverbed was recorded during
the flood peak by the hydrometer on the Misa River (i.e., 5.39 m as reported in Figure 2d). The height from the riverbed
to the base of the deck was estimated, and this value was subtracted from the maximum height measured by the
hydrometer, resulting in a MWL of 2.18 meters.
In the case of the Bellisio Solfare asset (Figure 4, panel f8), the MWL was estimated by considering the mud marks height
at the closest building on the hydrographic left of the Cesano River. The measured MWL at this building, used as a
reference, is 1.45 m. Thus, considering the DTM difference between the refinery and this site, the resulting MWL at
Bellisio Solfare is equal to 2.66 m.
### 4.1.2   Factors influencing flood damage
In this study, the following factors were considered as those that can potentially contribute to the damage to CH assets:
maximum water level outside the building (MWL), minimum distance between asset and river (ΔD), difference between
the elevation of CH asset and the elevation of the riverbed (ΔE); difference between DTM and filled DTM (ΔDTM),
average slope of the river (RS), local slope (LS), curvature (CU), Topographic Wetness Index (TWI), Terrain Ruggedness
Index (TRI).
The procedures described in Sect. 2.2.2 allowed us to investigate which factors contributed significantly to both the LTV
and $LIV$ of the CH assets. Among all the factors analyzed, RS, MWL, and ΔE showed some correlation to LTV (Figure
5a-c), while for all others the correlation proved to be negligible. The same trend resulted also correlating the $LIV$ with
the same contributing factors (Figure 5d-e). This can be explained as the $LIV$ is linked to the LTV. Indeed, if an asset is



destroyed, all the intangible values are lost too. Overall, there is a greater correlation between LTV and contributing
factors than *LIV*, as the aspects that are not strictly related to physical parameters are considered when assessing *LIV*.
The factors RS and LTV (Figure 5a), considering the 500 m stretch upstream of the single asset of a group of assets
(RS500), exhibit both a higher correlation and a lower dispersion ($R^2$=0.91, RMSE= 0.12). Also considering the 1000 m
stretch upstream from the CH (RS1000), the LTV-RS relationship is clear, although it results in a lower correlation and
greater dispersion ($R^2$=0.75, RMSE=0.15) than considering the RS500 factor. These results show that an increase in RS
corresponds to an increase in LTV. Both 500 m and 1000 m were considered as there are no clear recommendations in
the literature on whether the flow of a river adapts to the slope of the riverbed. Nevertheless, considering these distances,
it is reasonable to assume that the slope of the riverbed affects the energy of the flowing water and thus can be used as a
valid proxy for current velocity. As observed, the dynamics of the flood event were different throughout the basins (Sect.
3.2). In the case of the Misa River in Senigallia (RS500,1000=0.001 m/m), the flooding that occurred was mainly caused
by the overtopping of the 2 bridges present, which in turn caused a progressive and slow rise in water levels throughout
the city. This scenario resulted in damage to CH primarily attributable to water stagnation and the accumulation of fine
sediments (ranging from clays to sands), rather than the direct impact of hydrodynamic forces from flowing water. Indeed,
for all the CH assets, the minimum LTV (5) was observed (Table 4). The only exception is the Garibaldi Bridge, which
was more severely damaged (LTV=15) as it was obstructed due to the passage of woody debris and the related pressure
exerted on it. On the other hand, for the sites in the Burano and Cesano basins, a steeper slope caused greater damage due
to the hydrodynamic force of the water impacting the CH assets. This is evidenced by some videos recorded at Cantiano
(as described in Sect. 3.2), but especially by the destruction of the Bellisio Solfare refinery (LTV=30). In this case, the
slope of the Cesano River was sufficient to transport and deposit large amounts of floating and coarse debris, including
wood, gravel, and boulders, which contributed to the destruction of the site. However, it is also worth noting that this site
was in a poor state of conservation, that possibly reduced structural resistance.

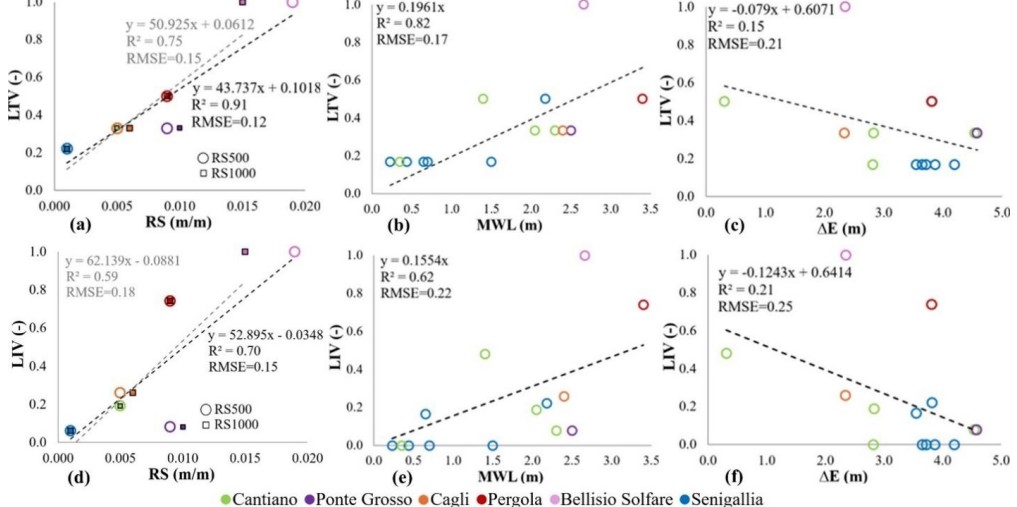


**Figure 5.** Relations between normalized LTV (a-c) and *LIV* (d-e) with influencing contributing factors: (a,d) RS, considering distances
of 500 m (black line) and 1000 m (grey line) upstream from the single asset or group of assets; (b,e) MWL relative to the ground floor
of each asset; (c,f) ΔE, the elevation difference between the asset and the riverbed.
As concerns the correlation between LTV and MWL, Figure 5b highlights a clear relationship. Namely, the higher the
flood depth, the greater the damage, as generally found in the literature for stage-damage functions. However, a lower




correlation is observed than the LTV-RS500 relationship as well as also a higher dispersion ($R^2$=0.82, RMSE=0.17). A
higher RMSE value can be justified by the Bellisio Solfare site, which represents an outlier. Indeed, the maximum
assigned LTV value due to its destruction is not solely linked to the MWL, but rather to the energy of the flow, as
demonstrated above.
The lowest correlation and the highest dispersion ($R^2$=0.15, RMSE=0.21) correspond to the LTV-$\Delta$E relationship (Figure
5c).
Overall, the following results are worth highlighting:
• The correlation between LTV and *LIV* with $\Delta$E is not statistically significant (p-value > 0.05).
• LTV and *LIV* are highly correlated (Pearson's R=0.93 and p-value < 0.05). Despite *LIV* considering factors not
directly related to the physical characteristics of a flood event, it still correlates well with LTV. Indeed, aesthetic
and communal value losses are generally sensitive to flood impacts, while evidential and historical values persist
despite flood damage, as the asset remains a testament to historical eras and past activities. However, if the asset
is destroyed, also intangible values are lost.
• RS (i.e., a proxy for river flow velocity) is highly correlated with LTV and *LIV* (Pearson's R=0.85 and 0.84,
respectively, and p-value < 0.05) but not significantly correlated with MWL (Pearson's R=0.62 and p-value >
0.05). Therefore, both RS and MWL are crucial for accurately estimating damage.
As mentioned in Sect. 2.2.4, also intrinsic factors can potentially influence the damages to CH. The presence of valuable
contents, especially if exposed at a low level with respect to the ground floor, increases the amount of damage, and then
the restoration cost. Indeed, the religious architectures that contain paintings, precious pews, and ancient elements such
as organs, have incurred in moderate or severe LTV, specifically the churches of S. Maria delle Tinte, S. Giovanni Battista,
and S. Nicolò (Table 4). On the other hand, although the S. Agostino and S. Maria del Porto churches contain artworks,
they have not experienced a loss in tangible value. This is attributed to their elevated positioning above ground floor level.
However, it could be noteworthy that their low LTV can also be attributed to their relatively low MWL (Table 4). A more
explanatory perspective on the positive impact of elevation on damage is the S. Nicolò church. Indeed, in this case, despite
a high MWL, the associated LTV is relatively low, as it is supra-elevated at 1.12 m above ground floor level (Table 4).
Even the state of conservation could influence the degree of damage. Indeed, the poor state of conservation reduced the
Bellisio Solfare asset capacity to resist the impact of the water and debris mixture, contributing to its destruction. This
data confirms that the degree of conservation can directly impact the extent of damage observed following a flood event
(Stephenson and D'Ayala, 2014; Salazar et al., 2024).
Studies in literature pinpoint the role of construction material in determining the vulnerability of CH assets (Balasbaneh
et al., 2020; Brokerhof et al., 2023). However, no relations were found for this parameter, as all the surveyed assets are
characterized by the same material (i.e., masonry structure). The only exception is the Ponte Garibaldi, which was
constructed with a reinforced concrete structure.
Among the factors that have contributed significantly to the overflowing of rivers during the 2022 Marche flood event
are bridges and culverts, which were clogged. In Cantiano, the inadequacy of the culverted section at the entrance of the
urban area resulted in insufficient drainage of the Burano River, leading to overflow and sediment deposition. In Pergola,
a bridge near the S. Maria delle Tinte church was blocked by sediment and woody debris, resulting in flooding of the
surrounding area. In Senigallia, large woody debris blocked Ponte Garibaldi, causing the flooding of the city. It is widely
observed that bridges and culverts can become clogged during intense bed load transport, hyper-concentrated flow, or
debris flow events, leading to massive overflows. To mitigate the risk of clogging in complex urban environments, a river
management approach that incorporates optimized design principles based on adequate field surveys, numerical



modelling, and laboratory experiments is desirable (Gschnitzer et al., 2017; Amaddii et al., 2022, 2023; Martín-Vide et
al., 2023; Zugliani et al., 2023). These measures would also positively impact the preservation of ancient CH assets, which
are now confronted with heightened flood risks due to climate change, a risk likely lower during their construction.
**4.2 Comparison between ex-post and ex-ante damage assessment**
In this section, the results obtained through the methodology outlined in Sect. 2.1 are presented and compared to the
results of the ex-post damage assessment, considering only the LTV.
The first issue with the flood hazard map is its low degree of detail. Indeed, all the areas investigated are in the same
class, namely "medium probability (low-frequency floods)", and the map lacks some useful information, such as water
height or velocity. Thus, assets can only be included or excluded from floodable areas. Overlapping the assets of the MIC
database with the official map of flood hazard areas, 55 potentially damaged assets were identified. These assets were
then categorized based on their typology into various damage classes: 41 are included at risk of very high damage, 6 as
high, 5 as medium, and 2 as low. One of the individuated assets ("Fiorentino Basso") remains unclassified due to
insufficient information available in the MIC database regarding its type. Additionally, the MIC database lacks
information regarding the type of value associated with each asset. It is noteworthy that only 5 in 55 identified assets are
listed as damaged cultural heritage in Table 4. Indeed, 37 cultural assets are residential, productive, rural, or tertiary
architectures, with no local or touristic/cultural interest. Moreover, some religious architectures or historical
infrastructures that are located in flood hazard areas were not damaged by the flood during the 15-16 September 2022
event.
In addition, it should be emphasized that 9 assets defy the ex-ante damage assessment, even if identified as damaged
during the field survey. This discrepancy arises either from their absence in the MIC database (such as Ponte Garibaldi,
S. Emidio oratory, and S. Maria del Porto church) or because they do not overlap with the flood hazard areas (including
Portici Ercolani, Bellisio Solfare refinery, Filanda Serica, historical buildings Via Fiorucci, S. Agostino church, and S.
Nicolò church).
These findings highlight the main issues with the MIC database:
● Some assets may be inaccurately located (e.g., Bellisio Solfare refinery).
● In cases where assets have an extended area and only a small portion is potentially inundated, the point shapefile
may not accurately represent their exposure, as it could be situated in unexposed areas (as observed with the
historical buildings Via Fiorucci and S. Agostino church). In the case of widespread assets or constructions with
a linear footprint (i.e., assets including several buildings along a road, or porches such as Portici Ercolani) only
one centroid point representative of the location exists.
Consequently, the comparison between the ex-ante and the ex-post damage assessments is feasible only for five assets:
Porta Lambertina, Ponte Grosso, Foro Annonario, S. Giovanni Collegiate, and S. Maria delle Tinte church. Consistently
with observations, from the ex-ante damage assessment it derives that the two churches fall in a very high damage class,
the Ponte Grosso bridge falls is in a medium damage class, and the open space Foro Annonario falls is in a low damage
class. Observed losses thus confirm that religious architectures are the most vulnerable to flooding as assumed in most of
the ex-ante flood risk assessment works in literature (Garrote et al., 2020; Arrighi et al., 2023). Concerning Porta
Lambertina, it resulted in a high damage class, while the ex-post assessment resulted in being slightly damaged, as only
mud marks were observed.





### 5. Conclusions

This paper developed an ex-post flood damage assessment method for CH assets. This yields a semi-quantitative on-site evaluation of losses (i.e., not in monetary terms), both in terms of intangible and tangible impacts, that based on the best of our knowledge constitutes a novel aspect. The method consists of four main steps: (i) identifying CH assets potentially damaged by the flood; (ii) collecting post-event field data, through an ad-hoc developed survey form; (iii) evaluating the losses in both intangible and tangible values; and (iv) analyzing the factors contributing to flood damage. For step (ii), it is crucial to visit the damaged sites as soon as possible to collect data and information that may become unavailable due to restoration work. The use of the proposed form allows a quick easy, and reproducible way for the post-event flood data evaluation aimed at the direct assessment of losses in intangible and tangible values to CH assets. Then, step (iii) allows us to estimate the level of losses caused by floods on both tangible and intangible values to different types of CH assets. Finally, the findings from step (iv) allow for a better understanding of the causative phenomena aimed at valuable insights for disaster risk management.

The method was applied to the CH assets damaged by the flood event that occurred on September 15-16 in the Burano, Cesano, and Misa basins (Marche Region, Italy). The main findings that can be drawn from the application of the proposed method are the following:

- Post-event field survey is fundamental for gathering data and information on the hazard characteristics, such as water depths, together with losses in intangible and tangible values and for subsequent analysis (e.g., GIS processing). Ex-post flood damage information for CH is relevant for verifying the hypothesis of existing methods based on expert judgement. Moreover, it poses the basis for developing empirical flood vulnerability functions for CH. Peculiarities of CH, such as raised floors, presence of valuable artworks, and state of conservation are found to be relevant for flood vulnerability. Thus, where this information is not available, on-site inspections are suggested to better characterize actual exposure and vulnerability for ex-ante risk analysis.

- The LTV is well correlated with the MWL, consistently with damages to other building types. Additionally, there is also a strong correlation between LTV and the average slope of the riverbed, considering both 500 m and 1000 m upstream of the assets. The slope of the riverbed, a proxy of river flow velocity, can thus be considered as one of the possible contributing damage factors (as the measured or estimated data of water velocity is difficult to obtain).

- The *LIV* correlates well to the same contributing factors, however, *LIV* data show a lower $R^2$ and a larger spread demonstrating that intangible aspects are less dependent on flood characteristics. Nevertheless, LTV and *LIV* are highly correlated, since some intangible values, e.g., aesthetic and communal values are sensitive to physical flood damage, e.g., lack of accessibility.

- RS (i.e., a proxy for river flow velocity) is highly correlated with LTV and *LIV* but not significantly correlated with MWL, and therefore, both RS and MWL are crucial for accurately estimating damage.

However, the method also presents some limitations:

- The baseline pre-disaster intangible value is obtained by combining four different typologies of value (aesthetic, historical, evidential, communal) making some assumptions to identify the criteria for assigning the level of value to each intangible aspect. Additional or alternative aspects, not currently accounted for, could influence the assignment of intangible value.

- The limited number of surveyed assets does not allow for statistically robust relationships with contributing factors. Indeed, other potential contributing factors could affect the observed damage (e.g., construction material).


The existing exposure and vulnerability models, such as those by Arrighi et al. (2023), provide reasonable initial predictions of potential damage to cultural heritage (CH). However, it should be emphasized that the available exposure data are incomplete and inadequate for identifying all the flood-exposed assets and their vulnerability, leading to inaccurate ex-ante damage assessments to CH, specifically:

- In the Burano, Cesano, and Misa basins, the official flood hazard map lacks the necessary detail to distinguish which assets may suffer low or high flood damage, as it does not provide information on flood magnitude, such as water depth and velocity.

- The MIC database includes immovable and movable assets encompassing those currently under protection, and also those under verification. Therefore, an on-site direct check, conducted in collaboration with local authorities, is always necessary to determine whether an asset qualifies as cultural heritage. Furthermore, the database does not offer any information to delineate the value of assets, and in some cases, they are not accurately geo-localized.

This paper underscores the importance of post-flood data collection and analysis. The proposed method serves as a starting point for such data collection. Nevertheless, future research should include diverse cultural and geographic contexts to improve accuracy, as the contributing factors can differently influence the observed damage. An open-source, comprehensive CH database documenting flood-related damages, asset features (e.g., construction type, and building material), and factors describing the event magnitude (e.g., maximum water level) is needed. Additionally, quantifying tangible damage in monetary terms should allow us to obtain a more robust evaluation of the damage to CH assets. Nonetheless, it requires collaboration with government institutions to share monetary data (e.g., restoration costs). These steps would enhance flood risk management for CH conservation and help develop robust damage prediction models.

*Data availability*. GIS data and ex-post damage survey form will be made available in a public repository after acceptance.

*Author contributions*. CA conceptualized the research idea; CA and CDL equally contributed to the planning of the on-site data collection and performed the measurements; CA, CDL, and MA developed the methodology; MA, CDL, and CA analyzed the data; MA performed GIS analysis; MA handled the data visualization; CA supervised the research activity; MA and CDL wrote the manuscript draft; CA reviewed and edited the manuscript.

*Competing interests*. The authors declare that they have no conflict of interest.

*Acknowledgements*. The authors express their gratitude to the working group "MARCHE 2022" (https://sites.google.com/view/misa2022/home-page) for their collaboration in the post-event data collection phase.

*Financial support*. This study was carried out within the RETURN Extended Partnership and received funding from the European Union Next–Generation EU (National Recovery and Resilience Plan – NRRP, Mission 4, Component 2, Investment 1.3 – D.D. 1243 2/8/2022, PE0000005).

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
