# Peer review of "Tangible and intangible ex-post assessment of flood-induced damages to cultural heritage"

_Natural Hazards and Earth System Sciences, 2024_

## Author Response (AR1)

**Replies to Reviewers' Comments**

We want to thank the Reviewers for the useful comments aimed at improving our paper.
In the following we replay to each of their comments (Reviewer comment "**RC**" – Author reply "*AR*").
All changes mentioned below refer to the line numbers of the revised version of the manuscript (MS) with track changes.

**Reviewer #1 (Julius Schlumberger)**

**RC**: In the manuscript 'Tangible and intangible ex-post assessment of flood-induced damages to cultural heritage', the authors present a method for post-event assessment of tangible and intangible damages for cultural heritage sites. They apply the method to a case study and discuss some findings through exploring correlations and discuss strengths and limitations of the method. The authors write very clearly and present the new, relevant and insightful method comprehensively and with good support from literature. I enjoyed reading this manuscript and want to congratulate the authors on this valuable contribution to the scientific literature.

The only question to the authors refers to the quantification system for LTV and Value Score and LIV. The authors use ranges for these (5-30; 10-3; 0-1) without much discussion on how they determined these values to be reasonable and robust. Adding a few sentences to elaborate on how these ranges were chosen would enhance the manuscript. Additionally, discussing the limitations of such an expert-judgment-based quantification framework in the discussion section would be beneficial.

*AR*: We thank Reviewer #1 (Julius Schlumberger) for his positive evaluation of the manuscript and for marking an important aspect to be improved, namely explain further in detail how the quantification system for the LTV and Value Score and LIV was determined and discuss our choice.

Concerning the LTV, the scoring of the investigated CH is based directly on the degree of damage observed in the post-event fieldwork. The adopted scale ranges from 5 to 30, where the minimum value is greater than 0 as the classification system is designed for those assets actually affected by the flood that has suffered damage, even if only slightly, so that cleaning is sufficient to restore them. We assumed a degree of damage that varies linearly for the first three classes, namely from "slightly damaged" to "severely damaged". On the other hand, in the case of asset destruction, the LTV score is double that of the "severely damaged" class, to emphasize the difference between a severely damaged site that can be repaired despite the high cost, and a lost site that cannot be restored. In the new version of the MS, we added more information on the LTV classes and related score assignment (lines 284-290).

The assessment of the LIV is based first on the definition of the Value Score (i.e., the Baseline Value "BV" in the paper). The evaluation of the BV partially follows the classification systems proposed by HE (2008) and Romão and Paupério (2021), with some modifications. Specifically, while Romão and Paupério (2021) proposed 6 classes based on the level of interest of the asset (ranging from international to local, or unknown), our paper adopts a simplified approach with 4 BV classes: high, moderate, limited, and unknown, with corresponding scores of 10, 7, 3, and 0, respectively (lines 234-258 and Table 2). Once the BV is determined, the LIV estimate is calculated by multiplying the BV by a corresponding factor "D" (ranging from 0 to 1), which is based on the degree of damage observed post-event. This approach results in a maximum LIV score of 40, higher than the possible maximum LTV score (30), to emphasize the unique significance of intangible aspects over tangible ones. In the new version of the MS, we added more information on the LIV classes and related score assignment (lines 290-299).

Moreover, in the section 2.2.4 of the "Materials and Methods" chapter, we added a sentence (lines 378-380) specifying that the correlation analysis between LTV and LIV with the contributing factors were performed normalizing LTV and LIV relative to their maximum values.

Although the proposed LTV and LIV damage classification systems are specifically designed to CH assets based on the outlined criteria, we recognize that they are inherently subjective and that the scores assigned to

each class are not directly linked to physical variables. However, these systems could be valuable for a rapid initial assessment of post-flood damage to CH assets, particularly in situations where monetary damage estimates are unavailable, as often happens.

Moreover, as we developed an expert-judgment-based quantification method, a discussion on how the correlations and the corresponding conclusions change as a function of considering different scales is provided in the new version of the MS (lines 694-708).

**Reviewer #2**

**RC1 -** The submitted manuscript addresses a particularly important topic in the field of risk management for cultural heritage. Moreover, since research dealing with the post-event assessment of damage and loss to cultural heritage is a topic underrepresented in the scientific literature, namely in terms of flood impacts, the presented study is an important addition to the scholarship in this field.

The manuscript is well written and presents the new proposed method comprehensively with adequate support from the existing literature. While the manuscript is generally good, the following comments include several suggestions and questions with the objective of enhancing clarity and providing additional information to the reader.

*AR1 -* Thank you for the comment. We are pleased that our paper could be considered a significant contribution to research on evaluating flood impacts on cultural heritage.

Below you can find our point-by-point replies to your comments.

**RC2 -** Line 43: the reference (Garrote et al., 2019) does not address cultural heritage. Consider replacing it by the following reference which addresses the development of heritage site-specific vulnerability curves for flood risk assessment:

Figueiredo, R., Romao, X., Paupério, E. (2021). Component-based flood vulnerability modelling for cultural heritage buildings. International Journal of Disaster Risk Reduction, 61, 102323.

*AR2 -* The reference has been replaced with the one suggested (line 43).

**RC3 -** Line 45-46: For completeness, consider adding the reference (Garrote et al., 2020) which proposes a method for regional flood risk assessment at heritage sites

*AR3 -* The reference (Garrote et al., 2020) has been added to the revised version of the MS (lines 48-49).

**RC4 -** Lines 86-87: In the sentence "The proposed method is applied to the case study of the flood event that interested the Marche Region (Central Italy) on 15-16 September 2022.", consider replacing the word "interested by "impacted".

*AR4 -* Thanks for your suggestion. The word "interested" has been replaced by the word "impacted" (line 90).

**RC5 -** Lines 138-143: The definition of what is a cultural heritage site in the context of the current study needs to be clarified. Up to this point it appears that only officially designed/listed/protected sites are considered (since the authors have mentioned the MIC database earlier). However, in this sub-section, other criteria are mentioned for considering a site a cultural heritage site (i.e. tourism or local significance). Does this mean that non officially designated/listed/protected sites are considered? Moreover, the authors refer that sites from the official MIC database could actually be excluded. Why were officially designed/listed/protected sites excluded unless they actually don't exist? Are the authors putting more trust in tourism websites and social media reviews than in the official cultural heritage listings? If so, why? This needs to be clearly explained to better understand the context of the current study. Moreover, please also indicate what are the cultural heritage listing/designation levels included in the MIC database (World Heritage, national, regional, provincial, municipal, etc).

*AR5 -* Thanks for this comment. In the new version of the MS, we added some sentences to clarify these aspects, and in the following, we summarize that.

The MIC database is used as a reference to identify cultural heritage assets (Section 2.1, lines 101-108). The MIC database contains movable and immovable assets under protection with declared cultural interest at national and World Heritage listing levels. In addition, it also includes assets older than 50 or 70 years under evaluation to verify their effective cultural interest. So, assets older than 70 years are (should be) automatically included in the MIC database, pending verification of their cultural significance. As a result, the MIC database includes many private houses or industrial structures older than 70 years old, that lack cultural significance or touristic interest.

In the context of the current study, cultural heritage sites are defined as immovable and movable assets characterized by aesthetic, historical, testimonial, municipal, and tourist value (as reported in Section 2.2.1, lines 134-136). Therefore, based on the assumed definition of cultural heritage, some assets were excluded from the MIC database and others were added during the post-event survey (as reported in Section 2.2.1, lines 149-156):

- The assets listed in the MIC database that are not mentioned by official tourism websites or reviewed on major platforms (e.g., TripAdvisor and Google), were excluded from our analysis. Indeed, we assume that these assets have limited cultural interest in the context of this paper.
- Some assets were added according to recommendations provided by local authorities during post-event field verification, since they hold the right to be listed but were not reported in the listing (see below).

In particular, during the post-disaster data collection, local authorities reported sites that are not listed in the MIC database (Ponte Garibaldi, year 1930; S. Emidio oratory, year 1781; and S. Maria del Porto church, year 1858) (as mentioned in the results section 4.1.1, at lines 493-497). For instance, despite the S. Emidio oratory is a modest structure without significant artworks, it holds considerable cultural value. Indeed, local authorities informed us that it was built in 1781 to honor Cagli's citizens who perished in a devastating earthquake. In addition, for many decades it has hosted the traditional St. Emidio celebration. In conclusion, the listing level of all the assets damaged by the 2022 flood analyzed in our study is of national significance.

**RC6 -** Lines 151-164: The measurement of "Max. water level outside the building" and its use for damage inference requires some clarifications. When selecting the reference level, it is not clear if the selected "flat area" corresponds to the ground level of the heritage construction. If it does not, i.e., if the reference level is the streel level and the ground level of the heritage construction is above/below the street level, shouldn't there be a measurement of the construction ground level that would be subtracted/added to the mud mark level? Moreover, the example of the bridge that is given assumes that the deck is leveled, but in case it is arched, upwards or downwards, what should be the reference level?

*AR6 –* Thanks for the comment. We realised that the description of the MWL parameter, as well as the identification of a "flat area", was not very clear. For this reason, in the new version of the MS (lines 179-213) we have clarified these aspects, and we also added for completeness the "*hg*", "*Δq*", and "*mwl*" parameters. Moreover, in Table 1, we also included a diagram useful for further elucidation related to the measurement of these parameters.

Specifically, in the section "Construction features" reported in Table 1, we added the following aspects:

- Presence of basement floors (considering your comment, **RC19**)
- "*hg*", namely the "Height of the inside ground floor", described in lines 180-181.
- "*Δq*" represents the difference in elevation between the ground level outside the considered CH asset and a reference point in a flat area (lines 181-182).

Concerning the section "Flood characteristics" of Table 1, we have clarified at lines 182–213 the definitions of the MWL and *mwl* parameters, as well as the methods used to determine them (as summarised below):

- "MWL" corresponds to the maximum water level outside the construction, measured between the ground floor where the asset is located and mud marks that were still visible at the time of the field survey. This is the parameter used for damage inference (as described in the "Results and discussion" section, lines 630-646).

- "*mwl*" corresponds to the maximum water level inside the construction. Due to the peculiar features of the CH constructions, it could be very different from the MWL depending on variations in "*hg*", which may be more or less complex, as in the case of religious buildings.

However, the cultural assets of the current study were easily identified and were mostly located in flat areas, therefore, even if $\Delta q$ was measured during the field campaign, it was negligible (lines 622-623). For these reasons, the MWL measurements of all CH assets are referred to the outside ground floor level of each CH asset.

In the case of a downwards or upwards-arched bridge, the reference level should correspond to the intrados and extrados, respectively. A clarification on these aspects has been added in the new version of the MS (lines 214-216).

According to these definitions, we modified panels (h) and (i) in Figure 4 and we also added the depiction of MWL measurements in Figure 5.

**RC7 -** Also, consider replacing "building" by "construction" in the survey form since it is also applicable to bridges (given the example that is referred when measuring the water level) and other types of construction (based on the case study).

*AR7 -* Thanks for your suggestion. The term "building" has been replaced by "construction" where necessary.

**RC8 -** Line 176-178: The reason for excluding the "no value" category is not well supported. Not all sites possess all these categories of values. Some might only have 3 of the 4 categories, for example. Assigning the "no value" level to a category states this clearly. On the other hand, the "unknown value" refers to a lack of knowledge on the part of the assessor about the level of value which should not be mixed with the "no value" category (which is what the authors actually do in Table 2 since several descriptions for the Unknown category indicate there is no value for that category). Given these conceptual differences, if the authors choose to aggregate these two categories, they need to clearly mention that it is a simplification and explain why this simplification is relevant for the purpose of their study.

*AR8 -* Thanks for the remark. Based on the available data, it could be challenging to distinguish between sites that lack certain categories of value and assets whose value in those categories is unknown. Initially, we decided to exclude the "no value" category to avoid categorizing a site as having "no value" without certainty. However, following your suggestion, we agree that it is clearer to combine assets with unknown value and those with no value into a single category, renamed as the "Unknown or No Value Class." This simplification should therefore allow sites with limited significance information to be located correctly, avoiding evaluation errors.

We have clarified these aspects in lines 228-233.

**RC9 -** Table 2: In the Aesthetic category, the High value means it is a valuable structure AND there are valuable artworks inside? This is not clear. In the Historical category, the time limits are not clear. I assume that 1800 and 1900 structures refers to 19th century and 20th century constructions (if so, please use the time ranges in centuries for clarity, if not, please clarify what you mean). Assuming these time ranges, a construction of less than 70 years can also be from the 20th century. How is this solved when scoring this category?

*AR9 -* Thanks for these useful considerations. The high value in the aesthetic category includes assets with valuable structure and valuable artworks inside. We clarified it in Table 2.

Referring to the historical categories, we changed the time ranges in centuries in Table 2, as you suggested. Moreover, initially, we assigned structures younger than 70 years old with "unknown historical value". Indeed, according to Italian law, these structures are automatically included in the MIC database, pending cultural verification. However, thanks to your comment, we realized that, as defined in the first version of the manuscript, the "Unknown value class" overlaps with the "Limited value class".

In Table 2, the "Limited value class" will encompass the first mid of 20th-century cultural assets, while the "Unknown value class" will include 2nd mid of 20th-century cultural constructions.

**RC10 -** Moreover, the scores for each type of value do not reflect any type of difference in terms of international/national/regional significance. Why? Are all the sites considered in the study in the same level of listing/protection (i.e. are all the sites nationally designated sites, or regionally designated sites, for example?)?

*AR10 -* Thank you for your comment on this aspect, which indeed required clarification. Based on the available information, we decided to make the class value classification system presented in Table 2 independent of the listing/protection level of the CH assets (lines 237-258). Moreover, all the sites considered in the study are nationally designated, both those included in the MIC, as well as those sites not included in the MIC but that are considered culturally significant according to the suggestions of local authorities, as they meet the characteristics defined by the national law governing cultural heritage (lines 493-497).

**RC11 -** Line 187: The authors need to be more precise and detailed when referring to loss in tangible values. It is referred that this category of losses is connected to physical damage and the costs of restoration. However, much of the repair and the restoration works in a cultural heritage site are often connected to restoring or minimizing the loss of aesthetic values (which is an intangible value). Moreover, restoring a wooden floor made with some sort of woodworking technique that is important from an historical point of view and that was damaged by the flood is also a way to minimize the loss in the historical intangible value. Therefore, the authors need to make clear that LTV refers only to the level of physical impact of the event and the comprehensiveness of the necessary repair/restoration actions, while impact to the intangible values which also depend on the direct effect of the event (see another comment below) is captured using the LIV.

*AR11 –* Thanks for your comment. We added some sentences to clarify the aspects linked to the losses in tangible and intangible values (lines 261-282).

**RC12 -** Line 189: Consider including walls and arches as load bearing elements since many traditional constructions in masonry will have these elements.

*AR12 –* Thanks for your suggestion, these elements were added at line 264.

**RC13 -** Line 191: Most losses in intangible value are caused by the direct impact of floods, not the indirect impacts. The physical damage can cause loss of aesthetic value, and a level of physical damage that includes the total destruction of the site will cause total (or near total) loss of historical and evidential value. Please correct the sentence.

*AR13 -* Thank you for this clarification. As mentioned in the previous reply to your comment (*AR11*). These aspects have been clarified as suggested (lines 261-282).

**RC14 -** Line 224: Please indicate what DTM means.

*AR14 -* We have specified that DTM corresponds to Digital Terrain Model (lines 331-332).

**RC15 -** Line 230: Please indicate what DEM means.

*AR15 -* We have specified that DEM corresponds to Digital Elevation Model (line 337).

**RC16 -** Line 235: It has been previously mentioned that the MIC database represents the cultural heritage sites as points, not as polygons. So how were the necessary polygons established?

*AR16 -* Thank you for this observation. The polygon shapefile of the buildings is generally available from national or regional databases. If polygonal shapefiles are not available, the shape of the assets can be digitalized based on sufficiently detailed topographic maps or aerial photos (lines 342-363). Furthermore, in section 3, the sources of the GIS data are indicated (lines 385-391).

**RC17 -** Section 4.1.1: The authors need to clarify if the 14 sites that are considered in the study are all the sites that were impacted by the flood or just a sample. In case it is the latter, please indicate why only a sample of the sites was considered. Moreover, given the comments made before regarding what is considered a cultural

heritage site in the context of the current study (about Lines 138-143), please indicate which of these sites are not officially designated/listed heritage sites, if any.

*AR17 -* Thanks for this comment. According to the definition of cultural heritage provided in lines 134–136, we declared at line 489 (section 4.1.1) that the 14 sites considered in the study are all the cultural heritage sites affected by the flood event that we are aware of, also thanks to the collaboration with local authorities.

**RC18 -** Section 4.1.2: The authors should provide some comments regarding the following aspects:
1) since both LTV and LIV are based on scores that involve qualitative scales, and since these scales could be different (e.g. for LIV, the authors do not use the scale that is proposed in (Romao et al, 2021) for scoring V, or LTV could be scored using a multiplicative scale instead of an additive scale), some comments are needed to discuss how the correlations and the corresponding conclusions could change as a function of considering different scales. A sensitivity analysis of these results with respect to the assumptions made in the scoring scales would be needed to strengthen these conclusions.

*AR18 -* Thanks for these remarks. First, in the new version of the MS, we have added more information on the criteria adopted for the evaluation of the LTV and LIV classes and related score assignment (lines 2835-299). Moreover, a discussion on how the correlations and the corresponding conclusions change as a function of considering different scales is provided in the new version of the MS (lines 694-708), and the data used for these analyses are reported below. Different values of LTV and V (for the calculation of LIV) were tested. In both cases, linear and non-linear scales were tested (Table 1 and Table 2 reported below). The related results (Table 3 and Table 4 reported below) generally demonstrate that using different scales of LTV and V, the results in terms of $R^2$ and RMSE are not very different from those obtained using the data proposed in our paper (as described in lines 706-708). The only exception is related to the LTV-MWL relationship, for which a stronger correlation was obtained in comparison with the results obtained in our paper, using both linear and nonlinear scale.

*Table 1 – LTV scale*

|  | LTV | | | |
| --- | --- | --- | --- | --- |
| Paper De Lucia et al., 2024 | 5 | 10 | 15 | 30 |
| LTV - linear variation | 5 | 10 | 15 | 20 |
| LTV - non-linear variation | 5 | 10 | 20 | 40 |

*Table 2 – V scale (for the calculation of LIV)*

|  | V score (for LIV calculation) | | | |
| --- | --- | --- | --- | --- |
| Paper De Lucia et al., 2024 | 0 | 3 | 7 | 10 |
| V - linear variation | 0 | 6.7 | 13.3 | 20 |
| V - non-linear variation | 0 | 3 | 12.5 | 20 |

*Table 3 – LTV results*

| LTV | | | |
| --- | --- | --- | --- |
| Factor | Analysis type | R2 | RMSE |
| RS500 | **Paper De Lucia et al., 2024** | **0.91** | **0.12** |
|  | LTV - linear variation | 0.89 | 0.18 |
|  | LTV - non-linear variation | 0.9 | 0.13 |
| RS1000 | **Paper De Lucia et al., 2024** | **0.75** | **0.15** |
|  | LTV - linear variation | 0.8 | 0.18 |
|  | LTV - non-linear variation | 0.74 | 0.16 |
| MWL | **Paper De Lucia et al., 2024** | **0.62** | **0.22** |
|  | LTV - linear variation | 0.88 | 0.22 |
|  | LTV - non-linear variation | 0.75 | 0.18 |
| ΔE | **Paper De Lucia et al., 2024** | **0.21** | **0.26** |
|  | LTV - linear variation | 0.16 | 0.25 |
|  | LTV - non-linear variation | 0.16 | 0.21 |

*Table 4 – LIV results*

| LIV | | | |
| --- | --- | --- | --- |
| Factor | Analysis type | R2 | RMSE |
| RS500 | **Paper De Lucia et al., 2024** | **0.7** | **0.15** |
|  | V - linear variation | 0.71 | 0.15 |
|  | V - non-linear variation | 0.67 | 0.15 |
| RS1000 | **Paper De Lucia et al., 2024** | **0.59** | **0.18** |
|  | V - linear variation | 0.6 | 0.18 |
|  | V - non-linear variation | 0.56 | 0.18 |
| MWL | **Paper De Lucia et al., 2024** | **0.62** | **0.22** |
|  | V - linear variation | 0.64 | 0.22 |
|  | V - non-linear variation | 0.59 | 0.22 |
| ΔE | **Paper De Lucia et al., 2024** | **0.21** | **0.26** |
|  | V - linear variation | 0.14 | 0.25 |
|  | V - non-linear variation | 0.15 | 0.25 |

**RC19 -** 2) how would the proposed process change if the constructions had basements that were flooded? The MWL might need to be increased but this does not necessarily mean that the corresponding LIV or LTV values would increase also. This might also have an impact on the correlations. Some discussion on this issue would be appreciated.

*AR19* – Thanks for this interesting question. In our opinion, the MWL would not necessarily increase in the case of a basement, as it represents the maximum water level reached around the damaged CH asset. Conversely, *mwl*, i.e. the maximum water level inside the building, could vary. Its variation and related changes in LTV and LIV may depend on how the basement is hydraulically connected to the upper floors and whether it is connected or not. We have discussed the possibility of a flooded basement in lines 709-721.

**RC20 -** Section 4.2: Some clarifications are needed regarding the following aspects:
1) Of the 14 heritage sites, 5 were in the MIC database and the floodable areas and 9 were not. Regarding these 9, 3 were not in MIC database. So why were they considered as heritage sites? This comes back to the previous question about the definition of what is a cultural heritage site in the current study.
2) Of the 55 heritage sites that were identified as potentially damageable by a flood, only 5 were considered by this study. Is it true that the remaining 50 were not damaged by the flood?
The authors refer that "37 cultural assets are residential, productive, rural, or tertiary architectures, with no local or touristic/cultural interest", but they don't mention that they were not damaged by the flood? So, were they damaged or not? If they were damaged, why were they not considered in this study (this is also related to the comment made regarding Section 4.1.1)? Also, the claim that they have "no cultural interest" is strange given they are in the MIC database. Please discuss and clarify this issue.
Finally, the remaining 13 sites are "religious architectures or historical infrastructures that are located in flood hazard areas" that were not damaged by the flood? Please confirm this issue also.

*AR20 -* Thanks for noting that there is no clear explanation of this aspect in this section.
First, as mentioned in a previous comment (*AR17*) and as explained in Section 2.2.2, in this paper we considered as cultural heritage also assets that are not included in the MIC database, but that have been reported by the local authorities, during the field survey. In addition, referring to the cultural heritage definition mentioned in lines 134-136, not all the assets listed in the MIC database are considered cultural heritage for the purposes of this study. Assets not mentioned by local authorities, official tourism websites or lacking reviews on major platforms were excluded (lines 149-156).
Starting from these considerations, and based on what is described in lines 755-761, we provided more clarity on the number of CH sites considered for the comparison between ex-post and ex-ante damage assessment in lines 761-767.
Specifically, we realized that the sites that do not meet the definition of cultural heritage are 38 and not 37 (we corrected this in the new version of the MS at line 763). Therefore, out of the 55 heritage sites, 38 were excluded as they do not align with the cultural heritage definition mentioned in this paper (lines 134-136). Indeed, most of them are named in the MIC database as "single-family cottage", suggesting that they are private residential building that are not accessible by local community or tourists. Since these 38 sites were excluded, no data were collected on them, and it is unknown whether they were affected by the flood.
Conversely, 5 of 55 heritage sites (Ponte Grosso, S. Giovanni Battista collegiate, S. Maria delle Tinte church, Porta Lambertina, and Foro Annonario) are cultural heritage assets, in accordance with the definition mentioned in lines 134-136, and they resulted damaged. Additionally, 11 sites, which include religious architectures, historical infrastructures, also meet the cultural heritage criteria considered in this paper. However, despite their location within flood hazard areas as indicated on the official hazard map, local authorities have declared these 11 sites undamaged. Lastly, one of the 55 assets ("Fiorentino Basso") remains unclassified due to insufficient information available in the MIC database regarding its type.
Summarizing, overlapping the MIC database assets with the flood hazard maps 55 assets were identified, and it results as follows:
- 38 assets are not classified as cultural heritage (based on the definition provided in lines 134-136).
- 11 assets are considered as cultural heritage but were not damaged by the flood.
- 5 assets are considered as cultural heritage and were damaged by the flood.

- 1 asset ("Fiorentino Basso") has not been classified because the MIC database does not provide any kind of information about its typology, and it was not found on its geographic coordinates.

**RC21 -** "Finally, the remaining 13 sites are "religious architectures or historical infrastructures that are located in flood hazard areas" that were not damaged by the flood?":

*AR21 -* The cultural heritage sites that were not damaged by the flood are 11 and not 13 because one of them was excluded ("Fiorentino Basso"), and another one has been reclassified as no cultural heritage.
These aspects are clarified in lines 755-767 (as described in the previous reply *AR20*).

**RC22 -** Figures: the quality of Figures 2 a, c, and d) needs to be improved, as well as that of Figures 5 a) to f)

*AR22 -* The resolution of Figures 2 and 5 has been increased.

**RC23 -** References: Many references need to be revised to be consistent with the journal's format.

*AR23 -* In the new version of the MS, all references have been revised and the style is now consistent with the journal's format.

---

## Author Response (AR2)

Response to Editorial Comment

**C.1** Minor edit as from the abstract only it is not completely clear what tangible (LTV) and intangible (LIV) are. There is no mentioning of abbreviation for loss of (in)tangible value.

**R.1** We amended the abstract in order to clarify the definition of the acronyms LIV and LTV.